# The small molecule ISRIB reverses the effects of eIF2α phosphorylation on translation and stress granule assembly

Carmela Sidrauski[1]*, Anna M McGeachy[2], Nicholas T Ingolia[2], Peter Walter[1]*

[1]Department of Biochemistry and Biophysics, Howard Hughes Medical Institute, University of California, San Francis, San Francisco, United States; [2]Department of Molecular and Cell Biology, University of California, Berkeley, Berkeley, United States

**Abstract** Previously, we identified ISRIB as a potent inhibitor of the integrated stress response (ISR) and showed that ISRIB makes cells resistant to the effects of eIF2α phosphorylation and enhances long-term memory in rodents (*Sidrauski et al., 2013*). Here, we show by genome-wide in vivo ribosome profiling that translation of a restricted subset of mRNAs is induced upon ISR activation. ISRIB substantially reversed the translational effects elicited by phosphorylation of eIF2α and induced no major changes in translation or mRNA levels in unstressed cells. eIF2α phosphorylation-induced stress granule (SG) formation was blocked by ISRIB. Strikingly, ISRIB addition to stressed cells with pre-formed SGs induced their rapid disassembly, liberating mRNAs into the actively translating pool. Restoration of mRNA translation and modulation of SG dynamics may be an effective treatment of neurodegenerative diseases characterized by eIF2α phosphorylation, SG formation, and cognitive loss.

## Introduction

Diverse cellular conditions activate an integrated stress response (ISR) that rapidly reduces overall protein synthesis while sustaining or enhancing translation of specific transcripts whose products support adaptive stress responses. The ISR is mediated by diverse stress-sensing kinases that converge on a common target, serine 51 in eukaryotic translation initiation factor alpha (eIF2α) eliciting both global and gene-specific translational effects (*Harding et al., 2003*; *Wek et al., 2006*). Mammalian genomes encode four eIF2α kinases that drive this response: PKR-like endoplasmic reticulum (ER) kinase (PERK) is activated by the accumulation of unfolded polypeptides in the lumen of the ER, general control non-derepressible 2 (GCN2) kinase by amino acid starvation and UV light, protein kinase RNA-activated (PKR) by viral infection, and heme-regulated eIF2α kinase (HRI) by heme deficiency and redox stress. The eIF2α kinase PERK is also part of the unfolded protein response (UPR). This intracellular stress signaling network is comprised of three ER-localized transmembrane sensors, IRE1, ATF6, and PERK, which initiate unique signaling cascades upon sensing an increase in unfolded proteins in the ER lumen (*Walter and Ron, 2011*; *Pavitt and Ron, 2012*).

The common mediator of the ISR, eIF2α, is a subunit of an essential translation initiation factor conserved throughout eukaryotes and archaea. The heterotrimeric eIF2 complex (composed of subunits α, β and γ) brings initiator methionyl tRNA (Met-tRNA$_i$) to translation initiation complexes and mediates start codon recognition. It binds GTP along with Met-tRNA$_i$ to form a ternary complex (eIF2-GTP-Met-tRNA$_i$) that assembles, along with the 40S ribosomal subunit and several other initiation factors, into the 43S pre-initiation complex (PIC). The 43S PIC is recruited to the 5′ methylguanine cap of an mRNA and scans the 5′UTR for an AUG initiation codon (*Hinnebusch and Lorsch, 2012*). Start site codon recognition triggers GTP hydrolysis and phosphate release, which is followed by release of

*For correspondence:
carmelas@me.com (CS); Peter.Walter@ucsf.edu (PW)

Competing interests: The authors declare that no competing interests exist.

eIF2 from the 40S subunit, allowing binding of the 60S ribosomal subunit to join. After these events, the elongation phase of protein synthesis ensues. To engage in a new round of initiation, the newly released eIF2 complex has to be re-loaded with GTP, a reaction catalyzed by its dedicated guanine nucleotide exchange factor (GEF), the heteropentameric eukaryotic initiation factor 2B (eIF2B). Phosphorylation of eIF2α does not directly affect its function in the PIC, but rather inhibits eIF2B, thereby depleting ternary complex and reducing translation initiation (*Krishnamoorthy et al., 2001*). eIF2B complex is limiting in cells, present in lower abundance than eIF2; a small amount of phospho-eIF2α therefore acts as a competitive inhibitor with dramatic effects on eIF2B activity. When eIF2B is inhibited and ternary complex is unavailable, the rate of translation initiation decreases.

Unimpaired elongation in the face of reduced initiation allows translating ribosomes to run off of their mRNAs, generating naked mRNAs that can then bind to RNA-binding proteins (RBPs) and form messenger ribonucleoproteins, which can further assemble into stress granules (SGs). These cytoplasmic, non-membrane bounded organelles contain translationally stalled and silent mRNAs, 40S ribosomal subunits and their associated pre-initiation factors and RBPs; these RBPs facilitate the nucleation and reversible aggregation of SGs through reversible, low-affinity protein–protein interactions mediated by their low complexity domains (*Buchan and Parker, 2009*; *Kedersha and Anderson, 2009*; *Kato et al., 2012*).

Paradoxically, under conditions of reduced ternary complex formation and protein synthesis, a group of mRNAs is translationally up-regulated. These mRNAs contain short upstream open reading frames (uORFs) in their 5′ UTRs, which are required for their ISR-responsive translational control (*Hinnebusch, 2005*; *Jackson et al., 2010*). These target transcripts include mammalian ATF4 (a cAMP response element binding transcription factor) and CHOP (a pro-apoptotic transcription factor) (*Harding et al., 2000*; *Vattem and Wek, 2004*; *Palam et al., 2011*). ATF4 regulates the transcription of many genes involved in metabolism and nutrient uptake and thus is a major regulator of the transcriptional changes that ensue upon eIF2α phosphorylation and ISR induction (*Harding et al., 2003*). Although activation of this cellular program can initially mitigate the stress and confer cytoprotection, persistent and severe stress and its associated reduction in protein synthesis and CHOP activation lead to apoptosis (*Tabas and Ron, 2011*; *Lu et al., 2014*).

In animals, the ISR has been implicated in diverse processes ranging from the regulation of insulin production to learning and memory. These effects were studied first using genetics by generating knock-out mice lacking individual eIF2α kinases as well as a knock-in of the non-phosphorylatable allele eIF2α^S51A (Eif2s1^S51A). Homozygous loss of eIF2α phosphorylation leads to perinatal death but heterozygous eIF2α^+/S51A animals, which have reduced levels of eIF2α phosphorylation, grow into healthy adults showing phenotypes that demonstrate the importance of translation initiation in establishment of long-term memories (*Scheuner et al., 2001*). Behavioral tests demonstrated that PKR^−/−, GCN2^−/− and eIF2α^+/S51A animals display enhanced memory consolidation in learning paradigms of light training (*Costa-Mattioli et al., 2005, 2007*; *Zhu et al., 2011*). Pharmacological modulation of eIF2α phosphorylation represented an important advance, allowing easier discrimination between developmental and acute effects of ISR reduction and circumventing the lethal phenotype of homozygous eIF2α^S51A/S51A. Recent work identified small molecules that modulate the ISR pathway at distinct steps: (1) kinase inhibitors that target PERK or PKR (*Jammi et al., 2003*; *Atkins et al., 2013*); (2) an activator of HRI (*Chen et al., 2011*); (3) salubrinal, an inhibitor of eIF2α phosphatases (*Boyce et al., 2005*); and (4) ISRIB (*Sidrauski et al., 2013*). By a yet unknown mechanism, ISRIB blunts the effects of eIF2α phosphorylation in cells and thus represents the first bona fide ISR inhibitor acting downstream of all eIF2α kinases.

Here, we show that ISRIB reverses comprehensively and specifically the effects of eIF2α phosphorylation. By profiling the genome-wide translational program downstream of the ISR, we present the application of ribosome profiling to the ISR in mammalian cells, which allowed us to identify and quantify the translational changes that take place upon its induction and ISRIB treatment. Moreover, live cell imaging revealed that ISRIB addition can trigger a remarkably fast dissolution of phospho-eIF2α-dependent SGs in stressed cells, restoring translation.

## Results

### Ribosome profiling of ER stress in mammalian cells

We used ribosome profiling to characterize translational changes induced by ER stress. Deep sequencing of ribosome-protected mRNA fragments provides global, quantitative measurements of

translation and reveals the precise location of ribosomes on each mRNA (*Ingolia et al., 2009*, *2011*). We triggered the UPR in HEK293T cells by treating them with tunicamycin (Tm), a toxin that blocks N-linked glycosylation of ER-resident proteins. We chose to analyze an early time point (1 hr) in order to focus on translational changes preceding the extensive transcriptional induction that takes place upon activation of the three branches of the UPR (for a time course of UPR induction, see *Figure 3—figure supplement 1* in 10.7554/eLife.00498). After 1 hr of Tm or mock treatment, we added cycloheximide (CHX) to arrest translating ribosomes, lysed the cells, and digested the extract with nuclease to degrade mRNAs not protected by ribosomes. In parallel, we isolated total mRNAs to monitor any changes in mRNA levels. Ribosome profiling data revealed a discrete subset of mRNAs that were translationally up- or down-regulated more than twofold after UPR induction (*Figure 1A*, above or below box) as seen by changes in abundance of ribosome-protected fragments (RPF) ('Ribo-Seq', y-axis) without corresponding changes in mRNA levels ('mRNA-Seq', x-axis). Data points representing statistically significant changes in expression between Tm-treated and untreated ('UT') samples are highlighted in black.

Consistent with the well-established presence of regulatory uORFs in their 5′-UTRs, this genome-wide analysis identified four previously extensively studied mRNAs that displayed significant translational upregulation: ATF4, ATF5, CHOP, and GADD34, (*Figure 1A*, colored pink). The mRNAs encoding the closely paralogous transcription factors ATF4 and ATF5 are known translational targets of the ISR (*Lu et al., 2004*; *Vattem and Wek, 2004*; *Zhou et al., 2008*). They contain two uORFs (the second one overlapping with the coding sequence [CDS]) that govern their enhanced translational efficiency. The mRNAs encoding the pro-apoptotic transcription factor CHOP and the regulatory subunit of the eIF2α phosphatase GADD34 were also significantly upregulated at the translational level. Although both CHOP and GADD34 are also known transcriptional targets of ATF4, we did not detect significant induction of their mRNAs at this early time point (*Figure 1A*, lack of displacement along x-axis) indicating that at the time point chosen our analysis exclusively reports on translational effects. CHOP and GADD34 mRNAs also contain uORFs that allow for translational regulation upon eIF2α phosphorylation (*Lee et al., 2009*; *Palam et al., 2011*).

We identified a total of 78 mRNAs whose translation changed significantly and substantially (more than twofold) upon ER stress in HEK293T cells (listed in *Figure 1—source data 2A*). GO term analysis revealed the involvement of these genes in diverse functions and several encode for proteins with entirely unknown functions. Besides the four known ISR translational targets described above, six mRNAs in the list contain previously mapped uORFs as validated by ribosome profiling in the presence of a translation initiation inhibitor to mark initiation sites (*Figure 1A*, colored green and *Figure 1—source data 2B*) (*Lee et al., 2012*). Whereas 5% of the non-significantly changed genes in the Tm sample contain previously identified AUG-initiated uORFs, 14% of genes in the list of ISR-translational targets contain uORFs, indicating a significant enrichment (p $\leq$ 0.003, chi-squared test with Yates correction).

A seventh and novel uORF-containing translational target of the ISR encodes SLC35A4, a putative nucleotide-sugar transporter (*Song, 2013*). It was recently shown that the longest uORF of SLC35A4 is indeed translated because peptides corresponding to the encoded polypeptide were found in a whole proteome mass spectrometry study (*Kim et al., 2014*). Analysis of RPFs in the uORFs of the SLC35A4 and ATF4 mRNAs revealed significant ribosome density, further confirming that these regulatory uORFs are normally translated (*Figure 1—figure supplement 1*). Due to the reduced mRNA expression levels of ATF5, CHOP, and GADD34 in the absence of stress or at early time-points of UPR activation, we did not analyze the RPFs or mRNA reads at specific locations along these genes, as the read numbers were low.

Interestingly, there was a slight reduction in translation of mRNAs encoding ribosomal proteins and translation elongation factors (*Figure 1—figure supplement 2*, panel A). The translation of this functionally related class of ~100 abundant mRNAs, which have a 5′ terminal oligopyrimidine (5′ TOP) motif, is controlled by the activity of the mTOR kinase (*Meyuhas, 2000*; *Tang et al., 2001*; *Hsieh et al., 2012*; *Thoreen et al., 2012*). The concerted changes that we observed in their translation upon UPR activation suggest that ER stress and eIF2α phosphorylation affects 5′ TOP translation in HEK293T cells.

## ISRIB substantially reduced the translational effects elicited by stress and eIF2α phosphorylation

To study the translational effects of the small molecule ISRIB at a genome-wide level, we analyzed changes in RPFs and mRNA levels after addition of the drug to both ER-stressed and unstressed cells.

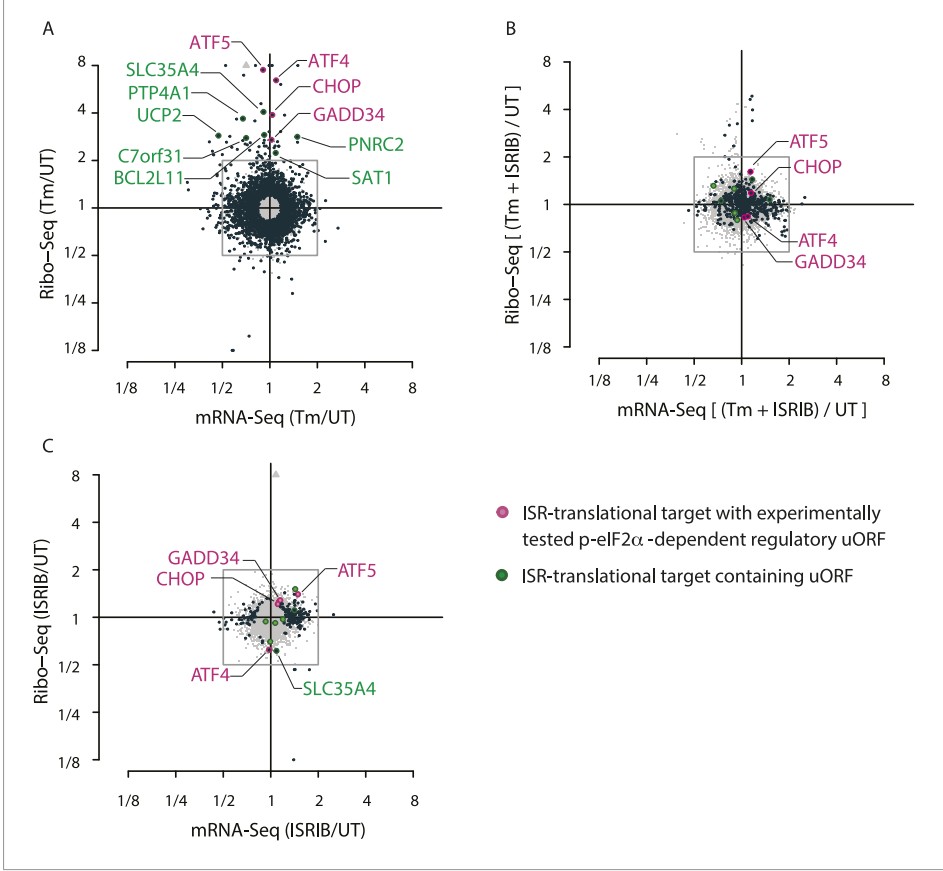

**Figure 1**. Translational regulation upon ER stress in mammalian cells. (**A**) Translational and mRNA changes in HEK293T cells upon ER stress. HEK293T cells were treated with or without 1 μg/ml of Tm for 1 hr. The y-axis represents fold changes in ribosome-protected fragments (Ribo-Seq) between Tm-treated and control samples. The x-axis represents fold changes in mRNA levels (mRNA-Seq) between Tm-treated and control samples. Data points reflecting significant changes (FDR-corrected p-value < 0.1) between Tm treated and untreated ('UT') samples are shown in black and non-significant changes are shown in light grey. Note that genes with significant changes (black circles) are numerous in Tm-treated cells and thus the cloud of genes with no significant changes (grey circles) is mostly hidden in the background. Genes with substantially enhanced RFPs and uORFs that are known to be phospho-eIF2α-dependently regulated are labeled pink. ISR-translational targets that contain previously identified uORFs are labeled in green. Triangles denote genes that fall beyond the axis range. The genes inside the grey box are those that change less than twofold in RPF or mRNA reads. *Figure 1—source data 2A* contains a list of all genes that change more than twofold in RPFs during Tm induction (FDR-corrected p-value < 0.1, corresponding to black circles above and below the box). (**B**) Translational and mRNA changes in cells co-treated with Tm and ISRIB. HEK293T cells were treated with or without 1 μg/ml of Tm and 200 nM ISRIB for 1 hr. The y-axis represents fold changes in ribosome-protected fragments (Ribo-Seq) between Tm + ISRIB-treated and control samples. The x-axis represents fold changes in mRNA levels (mRNA-Seq) between Tm + ISRIB-treated and control samples. Genes that significantly change when ISRIB co-administration modulates the effects of Tm treatment are shown in black (FDR-corrected p-value < 0.1). *Figure 1—source data 2C* contains a list of all genes that change more than twofold in RPFs during Tm and ISRIB treatment (FDR-corrected p-value < 0.1). The identity of the ISR-translational targets that contain previously identified uORFs (labeled in green) was not included in this panel as they all collapsed to the center of the plot. (**C**) Translational and mRNA changes in ISRIB-treated cells. HEK293T cells were treated with or without 200 nM ISRIB for 1 hr. The y-axis represents fold changes in ribosome-protected fragments (Ribo-Seq) between ISRIB-treated and control samples. The x-axis represents fold changes in mRNA levels (mRNA-Seq) between ISRIB-treated and control samples. Data points reflecting significant changes (FDR-corrected p-value < 0.1) between ISRIB-treated and untreated ('UT') samples are shown in black and non-significant changes are shown in light grey. *Figure 1—source data 2D* contains a list of all genes that change more than twofold in RPFs during ISRIB treatment (FDR-corrected p-value < 0.1, corresponding to black circles outside of the box). ATF4 and SLC35A4 (labeled in this panel) showed reduced translational efficiency upon addition of ISRIB. Two biological replicates were analyzed per condition. Number of reads aligned to the genome and ORFs for all samples are

*Figure 1. continued on next page*

*Figure 1. Continued*

found in *Figure 1—source data 2E*. Correlation plots for the replicates for each condition are found in *Figure 1—figure supplement 3*. mRNA abundance for all ORFs mapped are found in *Figure 1—figure supplement 4*. Read counts for all conditions and each individual transcript are found in *Figure 1—source data 1*. The Ribo-seq and mRNA-seq data have been deposited in NCBI's Gene Expression Omnibus and are accessible through GEO series accession number GSE65778.

The following source data and figure supplements are available for figure 1:

**Source data 1**. Read counts for all conditions and each individual transcript.

**Source data 2**. Source data for *Figure 1*.

**Figure supplement 1**. Ribosome and mRNA densities in the 5'UTR of ATF4 and SLC35A4.

**Figure supplement 2**. Translational regulation of mTOR targets upon ER-stress.

**Figure supplement 3**. Correlation plots for duplicate ribosome profiling experiments.

**Figure supplement 4**. Mean mRNA abundance of all genes mapped.

As seen in *Figure 1B*, ISRIB comprehensively blocked the translational changes that take place upon ER-stress. A large number of genes with a significant change in expression upon stress collapsed to the center of the plot with ISRIB and Tm co-treatment (*Figure 1B*, highlighted in black). Importantly, ISRIB abolished the induction of the known phospho-eIF2α-dependent translational targets (*Figure 1B*, colored pink) and the seven ISR-translational targets with previously identified uORFs (*Figure 1B*, colored green). The mRNAs that remained translationally induced in the presence of ISRIB are listed in *Figure 1—source data 2C*. In addition, ISRIB reversed the reduction in translation of mTOR target mRNAs upon ER stress (*Figure 1—figure supplement 2*, panel B).

Importantly, ISRIB treatment alone did not have general effects on translation in non-stressed cells, as revealed by the lack of substantial changes in RPFs in most cellular mRNAs, nor did it cause any significant changes in mRNA levels (*Figure 1C*) and mTOR target expression (*Figure 1—figure supplement 2*, panel C). In the absence of ER stress, ISRIB-treated cells behaved like untreated cells with the exception of a reduction in the basal level of translation of ATF4 and SLC35A4 mRNAs and a few additional mRNAs (*Figure 1C* and *Figure 1—source data 2C*). Taken together, these data strongly support the notion that ISRIB does not have global effects on translation, transcription, or mRNA stability in non-stressed cells and underscores its remarkable ability to counteract selectively the translational changes elicited by eIF2α phosphorylation in stressed cells.

## ISRIB prevents formation of stress granules exclusively triggered by eIF2α phosphorylation

Phosphorylation of eIF2α and reduction of ternary complex formation are tightly linked to the formation of stress granules (SGs) (*Kedersha et al., 2002*). ISRIB renders cells insensitive to the effects of eIF2α phosphorylation, thus leading to the prediction that it prevents SG formation as well. We tested this hypothesis by inducing SG formation using thapsigargin (Tg), a potent ER stressor that inhibits the ER calcium pump and was recently shown by ribosome profiling to yield analogous translational effects to tunicamycin (*Reid et al., 2014*). Microscopic detection of SGs required a stronger induction of ER stress than commonly achieved with Tm, making Tg the preferred inducer. We monitored SGs by performing immunofluorescence on eIF3a, a translation initiation factor that is recruited into SGs. As expected, we found that ISRIB significantly reduced their assembly upon co-treatment with Tg (*Figure 2A,B*). In addition, ISRIB prevented SG formation induced by arsenite (Ars), another widely used inducer of eIF2α phosphorylation via activation of HRI. As expected, both treatments induced eIF2α phosphorylation but only Tg induced the ER-resident kinase, PERK, as seen by its shift in mobility that is due to its extensive auto-phosphorylation (*Figure 2C*). Both the

PERK mobility shift and eIF2α phosphorylation elicited by Tg treatment were blocked by a PERK inhibitor (GSK707800; *Axten et al., 2012*) but not by ISRIB. Like ISRIB, and as expected by the block in eIF2α phosphorylation, the GSK PERK inhibitor prevented SG induction upon Tg addition (*Figure 2A*).

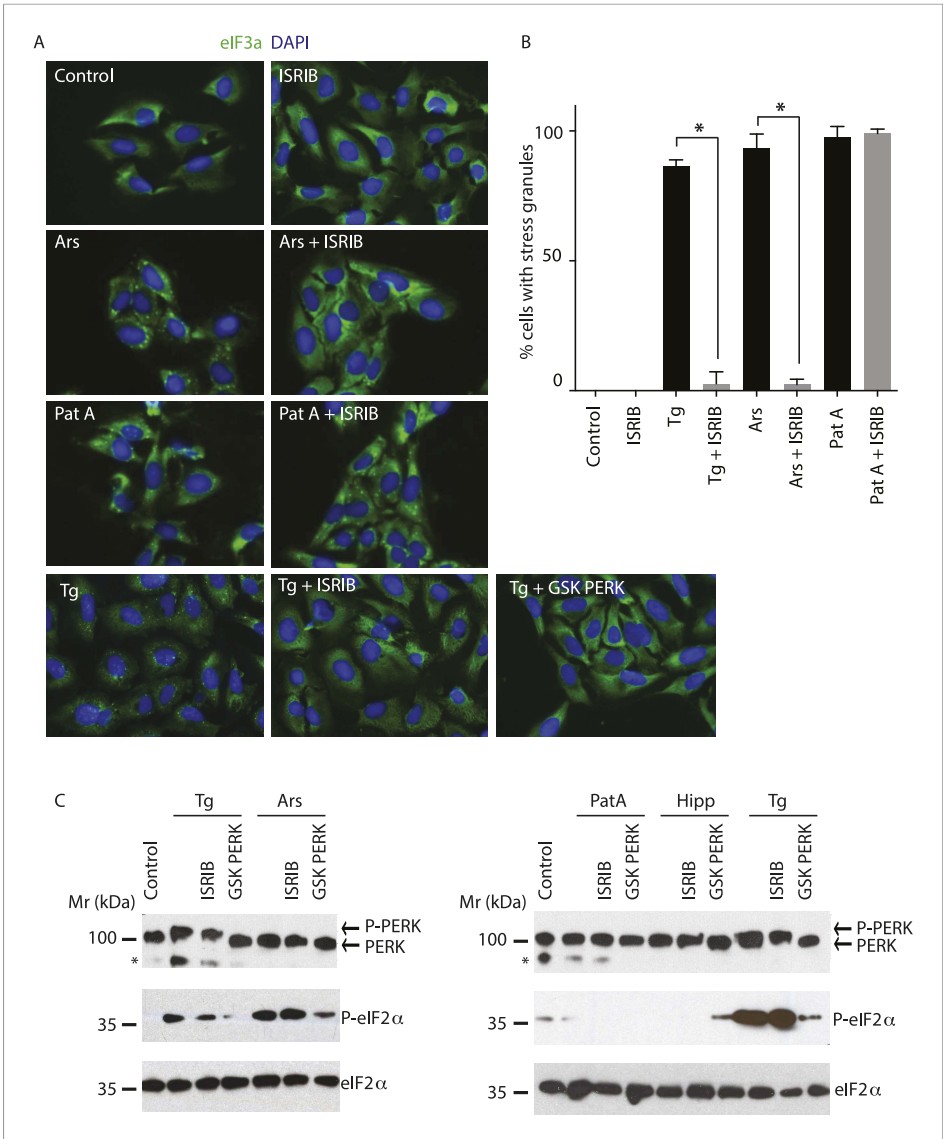

**Figure 2**. ISRIB blocks stress granule formation induced by eIF2α phosphorylation. (**A**) Immunofluorescence analysis (eIF3a) of U2OS cells treated with 200 nM Tg for 1 hr, 250 μM Ars for 30 min, or 100 nM Pat A for 30 min in the presence or absence of 200 nM ISRIB or 1 μM GSK797800 PERK inhibitor. A secondary Alexa Dye 488 anti-rabbit antibody was used to visualize eIF3a and DAPI was used to visualize nuclei. Representative images of at least two biological replicates are shown. (**B**) Quantitation of the percentage of cells containing stress granules in the different conditions described in **A**. Images were collected from at least two independent experiments and the number of cells with SGs or no SGs counted. The total number of cells counted for each condition was (sum of all replicates): Control (N = 81), ISRIB (N = 94), Tg (N = 122), Tg + ISRIB (N = 71), Ars (N = 85), Ars + ISRIB (N = 84), Pat A (N = 47) and Pat A + ISRIB (N = 64). No cells had SGs in Tg + PERK inh (N = 71). p-values are derived from a Student's *t*-test, *p < 0.05. (**C**) Immunoblot analysis of PERK, phospho eIF2α, and total eIF2α in U2OS cells treated as in **A**. Hippuristanol (Hipp) was used at 300 nM for 30 min. The right blot was overexposed to confirm the absence of induction of eIF2α phosphorylation upon Pat A and Hipp treatment. A representative blot of three independent experiments is shown. The asterisk (*) represents a background band or degradation product.

SG formation can also be induced in the absence of eIF2α phosphorylation by inhibiting the eIF4A helicase, which is part of the cap-binding eIF4F complex (*Mazroui et al., 2006*). Pateamine A (Pat A) binds to and inhibits this enzyme and blocks scanning of the PIC and translation initiation (*Dang et al., 2006*). In agreement, Pat A-induced SG formation but it did not cause eIF2α phosphorylation (*Figure 2A,C*). In contrast to phospho-eIF2α-induced SGs, Pat-A-induced SGs were not reduced by ISRIB (*Figure 2A,B*). Thus, ISRIB blocks phospho-eIF2α-dependent SG induction selectively.

## ISRIB triggers rapid disassembly of stress granules and restores translation

To visualize SG formation in living cells and to assess the effects of ISRIB on pre-formed SGs, we took advantage of a stable cell line expressing G3BP fused to GFP (*Kedersha et al., 2008*). In contrast to cell lines that overexpress SG-associated RNA binding proteins like G3BP, in this single clone-derived cell line, low expression of the fusion protein preserves stress-dependent regulation of SG assembly. We confirmed that in this cell line ISRIB significantly reduces SG formation driven by stresses that cause eIF2α phosphorylation (Tg and Ars) but not by phospho-eIF2α-independent induction through eIF4A inhibition (Pat A and hippuristanol [Hipp]) (*Figure 3A,B*) (*Cencic et al., 2012*). To match the strength of the stresses used in these experiments and minimize the toxic effects of these agents, we used the shortest incubation time and the lowest concentration of each stressor that resulted in SG formation in the majority of cells. ISRIB has an $EC_{50}$ of 5 nM as previously measured using an uORFs-ATF4-driven luciferase reporter in HEK293T cells (*Sidrauski et al., 2013*). In close agreement with the high potency measured in the reporter assay, ISRIB significantly reduced SG formation even at concentrations as low as 2 nM in U2OS cells and as expected, an inactive analog, ISRIB[inact], did not reduce their formation (*Figure 3—figure supplement 1*).

Treatment of cells with CHX disassembles SGs in the presence of ongoing stress (*Kedersha et al., 2000*; *Mollet et al., 2008*). This observation as well as other pharmacological and microscopy data revealed that SGs are highly dynamic structures with mRNAs quickly shuttling in and out. When these mRNAs leave SGs, translation is reinitiated; CHX then immobilizes elongating ribosomes and prevents mRNA re-entry into SGs. Because polyribosome disassembly is blocked by CHX yet required for SG assembly, CHX treatment dissolves pre-formed SGs. As seen in *Figure 3C*, a 10-min treatment with CHX following Tg induction of SGs (40 min) was sufficient to observe disassembly. Like CHX, ISRIB addition disassembled SGs within 10 min, even in the prolonged presence of the stressor Tg (*Figure 3C*). Whereas ISRIB restored translation of mRNAs that are liberated from SGs, as seen by the quick recovery in [$^{35}$S]-methionine incorporation, CHX further reduced protein synthesis (*Figure 3D* and *Figure 3—figure supplement 2*). These experiments demonstrate that ISRIB triggers disassembly of pre-formed SGs by loading dissociating mRNAs with actively translating ribosomes.

We next looked at the kinetics of SG disassembly upon ISRIB addition. Strikingly, after only 5 min of ISRIB treatment, Tg-induced SGs were no longer observed in cells (*Figure 3E* and *Video 1*). We also investigated the impact of ISRIB on P-bodies, a molecularly distinct class of RNA aggregates that serve as centers of mRNA decay (*Kedersha and Anderson, 2009*). The mRNA decay factor Dcp1 serves as a marker for these structures, and we visualized them in living cells using the fusion protein Dcp1-RFP. We saw that P-bodies were constitutively present in a percentage of the cells and were not affected by ISRIB treatment or by the stressors used to induce SGs over the time-course experiments explored here (*Figure 3E* red arrows, *Video 2* and data not shown).

## Discussion

ISRIB is the first reported antagonist of the ISR that blocks signaling downstream of all eIF2α kinases. It was shown to have good pharmacokinetic properties and brain penetration, making it a useful tool to study the systemic effects of acute inhibition of the pathway. We showed that ISRIB administration enhances long-term memory in rodents (*Sidrauski et al., 2013*). More recently, we showed by electrical recordings in brain slices that by preventing AMPAR down-regulation in the post-synaptic neuron, ISRIB blocks mGluR-mediated long-term depression (LTD), an effect that is dependent on eIF2α phosphorylation (*Di Prisco et al., 2014*). Comprehensive analyses of the cellular effects and kinetics of action of ISRIB are critical for interpretation of its in vivo effects and assessment of its therapeutic potential.

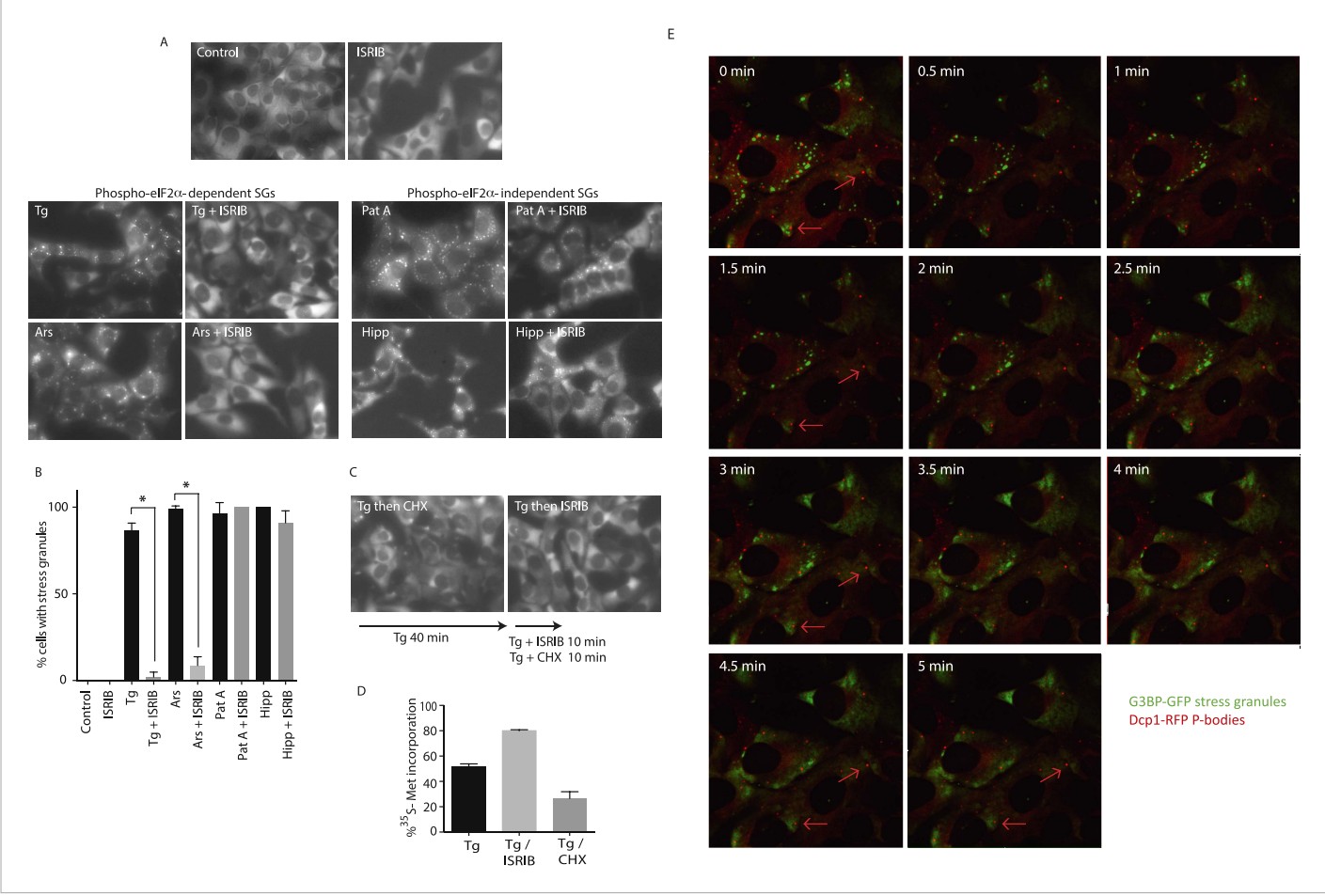

**Figure 3**. ISRIB addition rapidly dissolves pre-formed stress granules in live cells restoring translation. (**A**) Live cell imaging of stress granules in U2OS cells stably expressing G3BP-GFP (SG marker). Cells were treated with 200 nM Tg for 40 min, 250 μM Ars for 30 min, 100 nM Pat A for 30 min, or 300 nM Hipp in the presence or absence of 200 nM ISRIB. Cells were imaged using an epifluorescence microscope. Representative images of at least two biological replicates are shown. (**B**) Quantitation of the percentage of cells containing stress granules in the different conditions described in **A**. Images were collected from at least two independent experiments and the number of cells with SGs or no SGs counted. The number of cells analyzed for each condition was (sum of replicates): Control (N = 98), ISRIB (N = 81), Tg (N = 101), Tg + ISRIB (N = 84), Ars (N = 80), Ars + ISRIB (N = 55), Pat A (N = 58), Pat A + ISRIB (N = 50), Hipp (N = 41) and Hipp + ISRIB (N = 52). p-values are derived from a Student's t-test, *p < 0.05. (**C**) Stress granules were pre-formed with Tg for 40 min (as in *Figure 3A*) and then CHX (50 μg/ml) or ISRIB (200 nM) was added to the well, incubated for 10 min and images were collected. Representative images of at least two biological replicates are shown. (**D**) ISRIB quickly restores mRNA translation upon disassembly of stress granules. Cells were treated as in **C** with 200 nM Tg for 40 min and then DMSO, CHX (50 μg/ml), or ISRIB (200 nM) was added at the same time as [35S]-methionine. Cells were lysed after 15 min, protein was run in an SDS-PAGE gel and radioactivity was measured in each lane (N = 2, mean ± SD). (**E**) ISRIB quickly dissolves stress granules but does not affect P-bodies. Live cell imaging of U2OS cells stably expressing G3BP-GFP (SG marker) and Dcp1-RFP (P-body marker). Cells were treated with 200 nM Tg for 45 min followed by addition of 200 nM ISRIB at t = 0 min to the well and then imaged using spinning disk confocal microscopy. Images were collected every 30 s. The red arrows point to two representative P-bodies. Representative images of at least three biological replicates are shown.

The following figure supplements are available for figure 3:

**Figure supplement 1**. ISRIB dose response and inactive analog in stress granule assay.

**Figure supplement 2**. Representative SDS-PAGE gel of [35S]-methionine pulse as described in *Figure 3D*.

Our translational and transcriptional profiling confirmed that ISRIB treatment of ER-stressed cells substantially and comprehensively blocks the translational effects of eIF2α phosphorylation. ISRIB blocked SG formation that was triggered by eIF2α phosphorylation but did not abolish their assembly

upon eIF4A inhibition; eIF4A inhibitors do not cause eIF2α phosphorylation and can induce SGs in eIF2α$^{S51A/S51A}$ cells (*Mazroui et al., 2006*; *Mokas et al., 2009*). These data further support the notion that ISRIB solely inhibits cellular events that are a consequence of eIF2α phosphorylation. In agreement with these observations, we previously showed by polyribosome sedimentation analysis that ISRIB does not reverse bulk translational down-regulation triggered by inhibition of CAP-dependent initiation (*Sidrauski et al., 2013*). Moreover, ISRIB treatment alone did not induce overall changes in translation or mRNA levels. Taken together these data demonstrate that ISRIB is a pharmacological agent that acutely and specifically blocks the ISR and is thus an invaluable tool for in vivo studies.

## Translational regulation upon ISR induction

The method of ribosome profiling can monitor in vivo translation comprehensively and with nucleotide resolution (*Ingolia et al., 2009*). We used this method to monitor translation of all cellular mRNAs upon ISR activation. We found that a limited set of mRNAs is preferentially translated in a substantial manner upon a reduction in ternary complex assembly. Although previous large-scale analyses have revealed that almost 45% of all 5′ UTRs have at least one upstream uORF (*Calvo et al., 2009*; *Ingolia et al., 2011*), our data revealed that only a few of these mRNAs contain uORFs with regulatory properties that significantly enhance translation of their downstream coding sequences upon eIF2α phosphorylation. The canonical ISR translational targets, ATF4, ATF5, CHOP, and GADD34 mRNAs were significantly induced upon 1 hr treatment with the ER-stressor tunicamycin. The stress-induced, uORF-mediated regulation of GCN4 translation in yeast established the paradigm for this mode of regulation (*Dever et al., 1995*; *Grant et al., 1995*). As in mammalian cells, GCN2 is activated in amino acid-starved yeast by the accumulation of uncharged tRNAs, catalyzing eIF2α phosphorylation. The transcript encoding GCN4, a bZIP transcription factor with homology to mammalian ATF4, has four uORFs that modulate translation of its coding sequence upon stress. GCN4 induction is thought to occur via a re-scanning mechanism that allows 40S ribosomal subunits to remain mRNA-bound after completing the translation of short reading frames and subsequently reinitiate in the downstream coding sequence after reloading with ternary complex (*Hinnebusch, 2005*). The select mRNAs that are translationally upregulated in mammalian cells have uORFs that vary in number, length, and distance from the coding sequences. As was observed for GCN4, the uORF2 of ATF4 mRNA showed

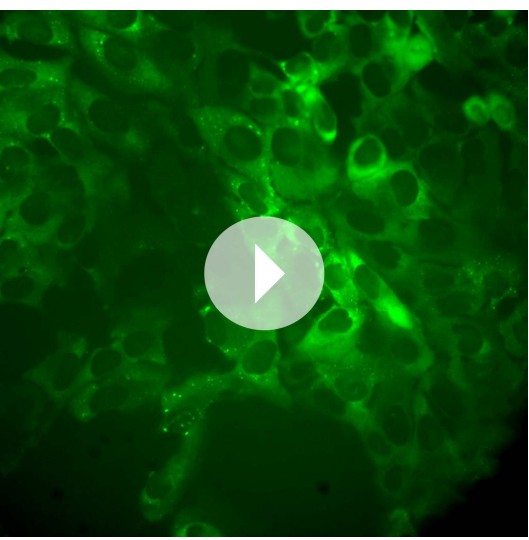

**Video 1.** ISRIB triggers stress granule disassembly. U2OS cells stably expressing G3BP-GFP (SG marker) and Dcp1-RFP (P-body marker) were treated with 200 nM Tg for 40 min and then 200 nM ISRIB was added at t = 0 min to the well and imaged using an epifluorescence microscope. Images of G3BP-GFP (SGs) were collected every 30 s.

ribosome density, supporting the notion that it is translated under normal growth conditions (*Ingolia et al., 2009*). Whether the same mechanism of rescanning is utilized by all these mRNAs is not known but, like ATF4, their regulation depends on their uORFs (*Vattem and Wek, 2004*; *Zhou et al., 2008*; *Lee et al., 2009*; *Palam et al., 2011*).

SLC35A4 is a novel translational target of the ISR. Ribosome profiling of HEK293T cells upon arsenite treatment, a potent inducer of eIF2α phosphorylation, also revealed the increased synthesis of SLC35A4 (*Andreev et al., 2015*). It belongs to a large family of nucleotide sugar transporters (NSTs) that are highly conserved transmembrane antiporters localized to the ER or Golgi apparatus (*Song, 2013*). The role of SLC35A4 in cells is unknown but it may function as the elusive ER-localized UDP-glucose transporter. This hypothesis is particularly attractive in the context of our data because unfolded ER proteins, which trigger the ISR, are continuously de- and re-glucosylated on their N-glycans using UDP-glucose as the glucose donor. Proteins with monoglucosylated N-glycans bind calnexin or calreticulin which promote protein folding. Translational induction of SLC35A4 may thus quickly enhance UDP-glucose transport into

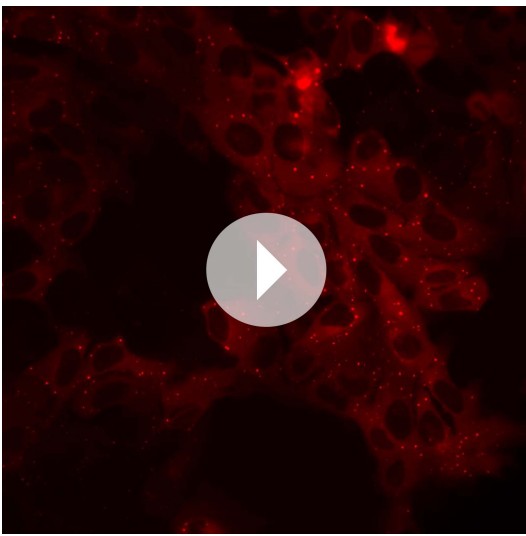

**Video 2.** ISRIB does not trigger disassembly of P-bodies. Images of Dcp1-RFP (P-bodies) corresponding to the same field of cells as in *Video 1* were collected every 30 s.

the ER lumen upon the accumulation of unfolded proteins in order to promote this pro-folding pathway.

Ribosome profiling upon activation of the UPR uncovered additional mRNAs induced upon eIF2α phosphorylation (our data, *Reid et al., 2014*; *Andreev et al., 2015*). Several of these mRNAs encode for proteins with entirely unknown functions and the remaining targets are involved in a wide range of cellular processes. Whether these ISR-induced translational targets are similarly regulated by the presence of uORFs in their 5′ UTRs remains to be determined with the construction of synthetic translational reporters. ISRIB blocked their differential translation, suggesting that these changes were due to phospho-eIF2α. There may be additional transcripts that are synthesized later during ISR activation, downstream of the early transcription factor targets such as ATF4, as well as tissue-specific mRNAs that are controlled by phospho-eIF2α. For example, OPHN1 is a neuron-specific mRNA containing uORFs that is translationally upregulated after mGluR engagement and eIF2α phosphorylation and induces LTD. By blocking the effects of phospho-eIF2α in cells, ISRIB also blocks mGluR-dependent LTD (*Di Prisco et al., 2014*). Ribosome profiling of glutamatergic neurons upon ISR induction may reveal additional transcripts whose translational control contributes to the molecular events underlying memory.

The ribosome profiling data presented here revealed that eIF2α phosphorylation modestly, yet significantly, decreased translation of a large number of ribosomal proteins and elongation factors. Although the decrease in translation of ribosomal proteins and elongation factors upon eIF2α phosphorylation is small in magnitude, its effects on bulk protein synthesis in the cell are significant as these represent a large number of highly expressed proteins. Translation of these mRNAs was previously shown to be under control of mTOR kinase, which regulates mRNA cap-binding factor eIF4E via phosphorylation of inhibitory eIF4E-binding proteins, thereby adjusting protein synthesis in cells in response to the cell's energy and nutrient status (*Ma and Blenis, 2009*). In this way, mTOR preferentially regulates translation of a group of mRNAs characterized by 5′ TOP motifs (*Hsieh et al., 2012*; *Thoreen et al., 2012*). Upon mTOR inhibition, translation of 5′ TOP mRNAs is reduced and the expression of factors required for protein synthesis is diminished.

The observed effect that eIF2α phosphorylation preferentially decreased translation of 5′ TOP mRNAs could, in principle, be due to inhibition of mTOR in response to ER stress. However, ISRIB reversed the translational changes, indicating that they are likely to be downstream consequences of eIF2α phosphorylation. Thus, if these translational changes do reflect altered mTOR activity, then the change in mTOR signaling must result from reduced translation mediated by eIF2α phosphorylation. Alternatively, eIF2α phosphorylation may lead to silencing of these mRNAs by recruiting them into SGs. The RNA binding proteins TIA-1 and TIAR, which are prominently SG-associated, were previously shown to bind to TOP mRNAs, leading to their translational downregulation upon amino acid starvation. This effect required both mTOR inhibition and GCN2 activation, the latter resulting in eIF2 phosphorylation (*Damgaard and Lykke-Andersen, 2011*). SGs also have been shown to recruit signaling molecules including upstream negative regulators of mTORC1 (raptor and DYRK3) and mTORC1 itself, and thus, SG formation may reduce their presence in the cytosol and impede translation of 5′ TOP mRNAs (*Thedieck et al., 2013*; *Wippich et al., 2013*).

## Stress granule dynamics and ISRIB

The dynamic nature of SGs allowed us to monitor the action of ISRIB upon its addition to live cells in real time. Strikingly, addition of ISRIB to stressed cells with pre-formed SGs lead to their quick dissolution (less than 5 min), liberating mRNAs back into the translational pool. A pulse of [35S]-

methionine confirmed the fast recovery in protein synthesis even in the presence of stress. Although the molecular target of ISRIB remains unknown, its quick action suggests a direct effect on translation initiation. Phospho-eIF2α resistance has been observed both in yeast and in mammalian cells. In yeast, mutations in eIF2B (the GEF for eIF2) and eIF5 (the 48S PIC-associated GTPase-activating protein for eIF2) have been reported to make cells insensitive to this phosphorylation event (*Vazquez de Aldana and Hinnebusch, 1994*; *Pavitt et al., 1997*, *1998*). In mammalian cells, TLR4 engagement in macrophages leads to increased eIF2B activity by removal of an inhibitory phosphorylation and insensitivity to ISR activation (*Woo et al., 2012*). Thus, ISRIB may directly or indirectly enhance the activity of eIF2B, eIF5, or other initiation factors, thus quickly reversing the cellular effects of phosphorylated eIF2α.

SGs contain a large number of RBPs that harbor low complexity sequence domains that nucleate through transient, low affinity interactions (*Kato et al., 2012*). These RBPs usually contain several RNA-binding domains and can associate with more than one mRNA; this multi-valency further favors the coalescence of RNA-protein granules. A conspicuous feature of some degenerative diseases is the cytoplasmic or nuclear aggregation of RBPs, driven in some cases by pathogenic mutations. TDP-43 and FUS mutations are found in amyotrophic lateral sclerosis (ALS) and frontotemporal lobar degeneration (FTLD) (*Li et al., 2013*), and mutations in hnRNPA1 and hnRPNPA2/B1 have also been found in ALS (*Kim et al., 2013*). Recent reports have also described the presence of RNA and RBPs in aggregates that form in prion disease, tauopathies, and Alzheimer's (*Vanderweyde et al., 2012*; *Ash et al., 2014*). The impact of these cytosolic aggregates on SG dynamics is not known, though they may hamper the ability of SGs to properly dissolve, thereby contributing to sustained translational attenuation and neurodegeneration. By quickly disassembling SGs even in the presence of stress, ISRIB may provide a useful therapeutic intervention in these diseases by antagonizing the cellular effects of pathogenic RNA-protein assemblies.

## Materials and methods

### Chemicals
Tunicamycin was obtained from Calbiochem EMB Bioscience. Thapsigargin, cycloheximide and sodium arsenite were obtained from Sigma–Aldrich. Hippuristanol and pateamine A were a kind gift from Jerry Pelletier. GSK797800 (PERK inhibitor) was obtained from TRC Inc. ISRIB (*Sidrauski et al., 2013*) and an inactive analog (754125) (*Di Prisco et al., 2014*) were synthesized in-house.

### Cell culture
HEK293T, U2OS, and U2OS GFP-G3BP/Dcp1-RFP cells were maintained at 37°C, 5% $CO_2$ in DMEM media supplemented with 10% FBS, L-glutamine and antibiotics (penicillin and streptomycin). U2OS cells stably expressing G3BP-GFP/Dcp1-RFP cells were a kind gift from Nancy Kedersha (*Kedersha et al., 2008*).

### Isolation of ribosome footprints and RNA
HEK293T cells were treated with or without 1 µg/ml of tunicamycin, tunicamycin and ISRIB (200 nM), or ISRIB for 1 hr. Cycloheximide (CHX) (100 µg/ml) was added for 2 min, cells were washed with ice cold PBS (with 100 µg/ml of CHX) and lysed in 20 mM Tris pH = 7.4 (RT), 200 mM NaCl, 15 mM MgCl, 1 mM DTT, 8% glycerol, 100 µg/ml CHX, 1% Triton and protease inhibitors (Roche complete EDTA-free). A syringe (25G5/8) was used to triturate cells, the lysate was clarified at 12,000 rpm for 10 min and half of the lysate was used for RNA extraction (Trizol, Invitrogen, Carlsbad, CA) and the other half was digested with RNase I (Ambion). The amount of RNase I and time of incubation was optimized for each sample based on the collapse of polyribosomes to the monosome peak as analyzed by analytical polyribosome gradients. The reaction was quenched with SUPERaseIn (Ambion, Life Technologies) and the digested lysate was then loaded on an 800 µl sucrose cushion (1.7 g of sucrose was dissolved in 3.9 ml of lysis buffer without Triton) and centrifuged in a TLA100.2 rotor at 70,000 rpm for 4 hr. The pellet was resuspended in 10 mM Tris pH = 7 (RT), and RNA was extracted (phenol/chloroform).

### Generation of sequencing libraries and data analysis
Sequencing libraries were generated as described in *Ingolia et al., 2012*. For data analysis, we used DESeq as described by *Anders and Huber (2010)*. P-adj values (p-values) were calculated using the R command 'p.adjust' for multiple comparisons and the BH method (Benjamini and Hochberg, 1995) to

correct for false discovery rate. The data in this publication have been deposited in NCBI's Gene Expression Omnibus and are accessible through GEO series accession number GSE65778.

## Immunofluorescence

U2OS cells were seeded on 4-well chamber slides (Lab-Tek) 18 hr prior to processing for immunofluorescence. Cells (80% confluent) were fixed with ice-cold methanol. The cells were then rinsed with PBS (Sigma) and blocked for 1 hr at room temperature in 0.5% BSA in PBS. The cells were then incubated overnight at 4°C with an anti-eIF3A rabbit antibody (#3411; Cell Signaling Technology) at a 1:1000 dilution in blocking buffer. The next morning the slides were washed three times (5 min each time) with PBS and then incubated for 1 hr at room temperature in a 1:1000 dilution (in 0.5% BSA in PBS) of secondary anti-rabbit antibody labeled with Alexa Dye 488 (Molecular Probes). The slides were washed three additional times with PBS. The slides were then mounted with antifade reagent with DAPI (Life Technologies P-36931). Lastly, the slides were imaged using a Zeiss Axiovert 200M epifluorescence microscope.

## Live cell microscopy

U2OS G3BP-GFP/Dcp1-RFP cells were plated in 8-well Lab-Tek chamber slides and switched to imaging media (lacking phenol red) upon addition of different stress inducers. Cells were either imaged using a Zeiss Axiovert 200M epifluorescence microscope or in a heated chamber using a spinning confocal epifluorescence microscope (Eclipse Ti-Nikon) and an Andor iXon3 camera.

## Protein analysis

Cells were washed with PBS and lysed in SDS-PAGE loading buffer (1% SDS, 62.5 mM Tris–HCl pH 6.8, 10% glycerol). Lysates were sonicated and loaded on Any-kD SDS-PAGE gels (BioRad). Proteins were transferred onto nitrocellulose and probed with primary antibodies diluted in Tris-buffered saline supplemented with 0.1% Tween 20 and 5% BSA. The following antibodies were used: PERK (D11A8) (1:1000), eIF2$\alpha$ (#9722; Cell Signaling technology) (1:1000), phospho-eIF2$\alpha$ (Ser51) (44728G; Invitrogen). An HRP-conjugated secondary antibody (Amersham) was employed to detect immune-reactive bands using enhanced chemiluminescence (SuperSignal, Thermo Scientific).

## [$^{35}$S]-methionine incorporation

U2OS GFP-G3BP/mRFP-DCP1a cells were seeded on 12-well plates, allowed to recover overnight and treated with 100 nM Tg for 40 min. ISRIB (200 nM) or CHX (50 μg/ml) was added at the same time as 50 μCi of [$^{35}$S]-methionine (Perkin Elmer) and incubated for 15 min. Cells were lysed by addition of SDS-PAGE loading buffer. Lysates were sonicated and equal amounts were loaded on SDS-PAGE gels (BioRad). The gel was dried and radioactive methionine incorporation was detected by exposure to a phosphor-screen and visualized with a Typhoon 9400 Variable Mode Imager (GE Healthcare).

## Acknowledgements

We thank Margaret Elvekrog, Voytek Okreglak, Shelley Starck, and Jirka Peschek for editing the manuscript and the members of the Walter lab for helpful discussions.

## Additional information

### Funding

| Funder | Grant reference number | Author |
|---|---|---|
| Howard Hughes Medical Institute (HHMI) | | Carmela Sidrauski, Peter Walter |
| Kinship Foundation | Searle Scholars Program 11-SSP-229 | Nicholas T Ingolia |
| National Institutes of Health (NIH) | T32GM007231 | Anna M McGeachy |

| Funder | Grant reference number | Author |
|---|---|---|
| Carnegie Institution of Washington | | Nicholas T Ingolia |

The funders had no role in study design, data collection and interpretation, or the decision to submit the work for publication.

## Author contributions

CS, NTI, Conception and design, Acquisition of data, Analysis and interpretation of data, Drafting or revising the article; AMMG, Acquisition of data, Analysis and interpretation of data, Drafting or revising the article; PW, Analysis and interpretation of data, Drafting or revising the article

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
