## [Decision Letter]

Thank you for sending your work entitled “The Small Molecule ISRIB Reverses eIF2α-phosphorylation-dependent Effects on Translation and Stress Granule Formation” for consideration at *eLife*. Your article has been evaluated by Randy Schekman (Senior editor) and three reviewers, one of whom is a member of our Board of Reviewing Editors.

The Reviewing editor and the other reviewers discussed their comments before we reached this decision, and the Reviewing editor has assembled the following comments to help you prepare a revised submission.

All three reviewers recognized the significance of your attempt to further characterize the consequences of ISRIB application to stressed cells and the potential for such a study to serve as an important advance. The methodology you chose to evaluate the impact on mRNA translation by ribosome profiling and on stress granule formation, by imaging, was likewise deemed appropriate to the task. However, the individual reviews and the consultative process that followed uncovered important problems concerning the interpretability of the data and the validity of the conclusions drawn. These problems are rather pervasive and it will not be possible for the Reviewing editor to decide alone if a revised version would be suited for publication. Thus, if you decide to revise your paper in accordance with the stipulations below, please bear in mind that it will need to be reviewed again by all three reviewers.

1) Detailed information concerning the Ribo-Seq experiments is missing: How many million mapped reads and how many replicates were done for each sample? The worry that data set may be inadequate is compounded by an impression that the read density for those elements of data that are presented in detail (Figure 1—figure supplement 1, for example) is rather low.

2) Significance figures for RNA transcript abundance are not reported.

3) The full power of single base resolution of the RPF analysis is not harnessed to de-convolute the effects of stress and ISRIB on the translation of uORFs in ATF4 and ATF5. It should be possible to assign reads to either the uORF or the main ORF even when they are overlapping as the reading frame is different.

4) The central conclusions of the paper, that ISRIB eliminated the effects of stress on translational regulation and that ISRIB is equivalent to mutations that abolish the ISR are not supported by the data. Numerous mRNA are differentially engaged by ribosomes in stressed & ISRIB treated sample(s) (Figure 2) compared to the stressed PERK^-/-^ cells (the benchmark used here for an ISR-inhibited system, Figure 2). These experiments may be further confounded by a methodological issue in that the cells shown in Figure 2 were apparently treated with tunicamycin for 1 hour, whereas those in Figure 2 for 30 minutes. These issue needs to be dealt with in some detail.

5) The effects of ISRIB on stress granule formation in the images shown seem convincing. But these need to be quantified and analyzed with appropriate statistical tools.

The reviewers’ individual comments are noted below.

Reviewer #1

This manuscript provides new experiments building on those published in a 2013 *eLife* article concerning the mechanism of action of an ISR inhibitor compounds called ISRIB.

The authors use complimentary techniques to those used in their original study to provide additional support to their model that ISRIB acts to nullify the downstream signalling events following induction of eIF2 phosphorylation. Two experimental strategies are employed here: 1) Ribosome profiling (Ribo-Seq) of cells to examine global translational control responses to ISR and its inhibition and 2) cell imaging via immunofluorescence and also live-cell GFP to examine the appearance of stress granules and P bodies-markers of cellular stress. Both of these analyses appear to confirm and extend the findings reported in the original study, but do not yet identify the mechanism by which the inhibitor ISRIB nullifies the ISR.

My major concerns are with the robustness of the data reported. In its present version this is not possible to fully assess.

1) For RNA and Ribo-Seq experiments, there is no information given concerning the depth of sequencing. How many million mapped reads and how many replicates were done for each sample?

2) The significant changes in ribosome occupancy are reported in supplementary tables, but changes in RNA transcript abundance are not reported. This data should also be added to the manuscript.

3) For the Ribo-Seq fold changes reported, how are reads mapping to overlapping ORFs handled? Eg Figure 1—figure supplement 1, shows reads covering the uORFs (green) and main ORFs (blue) that overlap.

4) Is the read density data in Figure 1—figure supplement 1 reporting a single representative sample, total summed reads from replicates (if done) or mean reads from multiple replicates? The read density is not very high for any of the genes shown, except for ATF4.

5) Can the authors confirm and comment on the presence of RPFs between the annotated uORF and main ORF for CHOP mRNA?

6) One perhaps unexpected finding was an apparent increase in uORF RPFs following stress. The authors have chosen not to elaborate on this point. The usual translational control models depicted often show an all or nothing extreme response, but it is not surprising at all that there are still ribosomes on uORFs that may get skipped more frequently under stress conditions. Perhaps an important measure of control is the proportion of ribosomes that skip uORF2 in ATF4 and 5. This could be quantified as the Ribo-Seq read data for the overlapping ORF regions should be in separate reading frames. It should be possible to assign reads to either the uORF or the main ORF. As a measure of stress mediated translational control, the authors could quantify the number and proportion of RPF within the ORFs key for translational control. For ATF4 and ATF5 ORF/uORF2 reads would give an alternative readout of the relative translational control.

7) Blots showing eIF2 phosphorylation relative to total eIF2 should be in each type of cell used (plus minus) treatment for the time points used in the Ribo Seq study. This could be added to the ATF4 blot which uses different time points.

8) For the immunofluorescence experiments and for the GFP experiments in Figure 3 and 4 and Figure 4–figure supplement 1, each experiment requires quantification of a larger number of cells than are shown in each representative image. For example in 100 cells in each of three separate experiments what proportion of cells (plus minus) error contain stress granules or P bodies as appropriate. Such an analysis would enable statistical treatment of the significance of the observations made.

9) What is the effect of the ISR and ISRIB on the other arms of the UPR at the times used here. Are sampling times too early to observe XBP1 splicing and changes in ribosome binding? If they are this should be stated, if they are not it would helpful to show XBP1 reads.

Reviewer #2

In this manuscript, the authors study the role of ISRIB, a small molecule targeting the integrated stress response (ISR) in two experimental systems: Ribosome profiling and stress-granules formation. The study of the ISR by ribosome profiling essentially confirms earlier studies. The effect of ISRIB and stress granules is surprising and novel, but the underlying mechanisms remain unknown. While the study is potentially interesting, there are some important issues to address.

Major issues:

Throughout the manuscript, there is a disconnection between the data and the conclusions. I will only highlight the major ones here:

1) One of the central conclusions of the paper is that ISRIB eliminated the effects of eIF2α phosphorylation. The authors wrote: “The translational output of ISRIB-treated cells to ER stress was remarkably similar to that of cells with a genetically ablated ISR”. This contradicts the data. In fact, there are major differences between the two data sets (Figure 2). I recommend providing a complete list of genes that change after ISRIB treatment and to study these changes, as it may shed light on ISRIB's function and/or target.

Supporting the idea that the changes caused by ISRIB (Figure 2) are actually tractable, ISRIB caused more changes (Figure 2) than Tm (Figure 1). The secret of ISRIB's function may be hidden in this dataset. This needs to be documented and exploited.

A related issue: The authors conclude that “ISRIB does not have general off-targets effects on translation, transcription or mRNA stability.”

Again, this contradicts the data. Beside, “off-target” is a not suitable here, since we don't know what is the target of ISRIB.

2) Figure 3: What are the vertical lines on PERK blots? Between first and second lane in the top blot and between the 9th and 10th lane in the bottom blot.

3) Figure 3: It would be good to see images with better resolution to appreciate the localization of eIF3α, which looks very interesting. I still see some stress-granules on Ars+ISRIB in some cells but in most cells, the signal is too strong to be analyzed. It would be good to see images of better resolution and some quantitative assessment of the effects.

4) The disappearance of stress-granules is the impressive finding of the manuscript but how does this all happen? It is difficult to follow what might be going on because of the lack of consistency in the conditions used and because adequate controls are not presented. From this paper and the previous one, it would appear that ISRIB acts upstream of ATF4 translation and is dependent upon eIF2α-P (but without affecting the latter)? This needs to be explored further to get some understanding of what is happening.

Figure 4C: It would be good to present gels to show the effects, as the authors have the images already. It would be interesting to appreciate the qualitative changes in translation. Does this match, what is seen by ribosome profiling?

5) The Discussion is very broad and not connected to the current dataset. The first sentence of the Discussion is incorrect, as there are other antagonists of the ISR. I suggest two options for a revised discussion: Either to focus the discussion on the current dataset or to provide additional data inline with the discussion (mTOR signaling and the ISR, memory and stress granules, memory and ribosome profiling data).

6) Figure 1: I didn't understand this panel. It needs to be clarified.

Reviewer #3

This study provides an important extension validating the compound ISRIB as a highly specific inhibitor of the canonical eIF2α-phosphorylation-dependent Integrated Stress Response. Given that original report was published in *eLife*, this present paper is most suitable as a linked research advance.

The most important finding is that the effects of ISRIB on mRNA translation, revealed by the unbiased tool of ribosome foot-print profiling and mRNA sequencing, are limited to effacement of the translational induction of a small set (five in total) of mRNAs that are translationally upregulated by the ISR. In this regard, ISRIB discretely mimics the effects of mutations that block eIF2α phosphorylation, either by interfering with the action of the upstream kinase (PERK-KO) or by precluding substrate phosphorylation in cis (eIF2α^S51A^).

Further support for a comprehensive defect in the ISR, introduced by ISRIB, is provided by evidence that stress granules (these are mysterious collections of mRNA binding proteins and translation factors that assemble in cells experiencing high levels of phosphorylated eIF2α) are rapidly disassembled by ISRIB.

In passing, this paper makes an important contribution to the study of the ISR: By confirming that its known positively regulated targets, cobbled together from biased searches (ATF4, ATF5, CHOP, GADD34), comprise a nearly complete list; the single newcomer being SLC35A. And by drawing attention to the fact that their translational induction by the ISR does not entail the loss of footprints on the short repressive uORFs (as predicted by the regulated translation re-initiation model).

An important caveat to these complementary comments is that this reviewer lacks the expertise to judge the technical validity of the RNA seq and ribosome footprinting and defers to other reviewers' expertise in this regard.

The lack of any measureable effect of ISRIB on baseline translation is surprising as there is evidence for basal activity of the ISR: For example PERK_KO, eIF2a_S51A and ATF4_KO cells all share a strong baseline requirement for amino acid supplementation (likely indicating that the ISR contributes to baseline ATF4-mediated gene expression). The authors may wish to comment on this point.

---

## [Author Response]

*All three reviewers recognized the significance of your attempt to further characterize the consequences of ISRIB application to stressed cells and the potential for such a study to serve as an important advance. The methodology you chose to evaluate the impact on mRNA translation by ribosome profiling and on stress granule formation, by imaging, was likewise deemed appropriate to the task. However, the individual reviews and the consultative process that followed uncovered important problems concerning the interpretability of the data and the validity of the conclusions drawn. These problems are rather pervasive and it will not be possible for the Reviewing editor to decide alone if a revised version would be suited for publication. Thus, if you decide to revise your paper in accordance with the stipulations below, please bear in mind that it will need to be reviewed again by all three reviewers*.

By now delivering a more focused message that directly relates to the parent *eLife* paper, we feel that these changes substantially improve the clarity and quality of this contribution and render it more appropriate as a Research advance.

We have made the following major changes in the manuscript:

1) To align the paper better with the previous publication, we excluded the Ribo-Seq and mRNA-seq data generated in mouse embryonic fibroblasts. Due to the intrinsic differences in kinetics and degree of UPR induction between MEFs and HEK293T cells, we now focus only on the ribosome profiling data generated in the latter. As reported in our previous *eLife* manuscript (10.7554/eLife.00498), in HEK293T cells, we observe a complete block both in ATF4 translational induction and bulk translational down regulation upon co-treatment with ISRIB, making it our cell line of choice to study the genome-wide effects of ISRIB. As requested by the reviewers, we include biological replicates for both Ribo-Seq and paired mRNA seq-data in all conditions reported in this cell type. Correlation coefficients between replicates and mRNA abundance for all genes are provided in new supplementary figures.

2) We included uORF ribosome occupancy for only the two highest expressed ISR-translational targets for which we obtained sufficient coverage in reads for both mRNA and Ribo-Seq data. We removed the uORF ribosome occupancy upon ER-stress plots as new insights pertaining to the mechanism of translation and rescanning of the uORFs in translationally stress-induced genes will require substantial further experimentation and would extend beyond the scope of this work. The primary goal of this Research Advance is to further characterize the biological effects of the small molecule ISRIB.

3) We have quantitated all stress granule data.

As requested, we have reworded the title to read “The Small Molecule ISRIB Reverses the Effects of eIF2α Phosphorylation on Translation and Stress Granule Assembly”.

*1) Detailed information concerning the Ribo-Seq experiments is missing: How many million mapped reads and how many replicates were done for each sample? The worry that data set may be inadequate is compounded by an impression that the read density for those elements of data that are presented in detail (*Figure 1—figure supplement 1*, for example) is rather low*.

The Ribo-Seq and mRNA-seq data in HEK293T cells was performed in duplicate (biological replicates) for each condition. Figure 1*–*figure supplement 7 contains the number of reads, number of reads mapped and the number of reads mapped to ORFs for each replicate sample. We will prepare a GEO submission for all datasets.

We have now included only ATF4 and SLC35A4, the two highest expressed ISR-targets, in our ribosome footprinting and mRNA read density plots along the genes (now Figure 1—figure supplement 3), for which we have sufficient coverage.

*2) Significance figures for RNA transcript abundance are not reported*.

We have included mean normalized counts for all transcripts in all conditions tested in Figure 1–figure supplement 9.

*3) The full power of single base resolution of the RPF analysis is not harnessed to de-convolute the effects of stress and ISRIB on the translation of uORFs in ATF4 and ATF5. It should be possible to assign reads to either the uORF or the main ORF even when they are overlapping as the reading frame is different*.

Due to the fact that all Ribo-Seq data in this manuscript was generated using cycloheximide to freeze translating ribososomes, and this may lead to an artifactual increase in 5’UTR ribosome occupancy, we have not quantitatively analyzed changes in uORF occupancy. Carefully de-convoluting uORF occupancy upon stress and ISRIB treatment will require further experimentation (including the use of a translation initiation inhibitor treatment to generate libraries). This analysis extends beyond the scope of the current Research Advance.

*4) The central conclusions of the paper, that ISRIB eliminated the effects of stress on translational regulation and that ISRIB is equivalent to mutations that abolish the ISR are not supported by the data. Numerous mRNA are differentially engaged by ribosomes in stressed & ISRIB treated sample(s) (*Figure 2*) compared to the stressed PERK*^*-/-*^
*cells (the benchmark used here for an ISR-inhibited system,*
Figure 2*). These experiments may be further confounded by a methodological issue in that the cells shown in*
Figure 2
*were apparently treated with tunicamycin for 1 hour, whereas those in*
Figure 2
*for 30 minutes. These issue needs to be dealt with in some detail*.

We have excluded the MEF data, and, because of differences observed between cell lines, we no longer benchmark PERK^-/-^ or eIF2α as an ISR-inhibited system. Importantly, we show that ISRIB comprehensively blocks translational upregulation of the vast majority of the mRNAs in HEK293T cells, including the well-established ISR translational targets ATF4, ATF5, CHOP and GADD34. Both the previously submitted manuscript as well as the current one contain tables that list for each condition the translationally upregulated mRNAs.

*5) The effects of ISRIB on stress granule formation in the images shown seem convincing. But these need to be quantified and analyzed with appropriate statistical tools*.

We have quantified all stress granule data and added the corresponding quantitation panels to Figures 2 and 3 with their corresponding statistics in each figure legend.

*The reviewers’ individual comments are noted below*.

Reviewer #1

*[…] My major concerns are with the robustness of the data reported. In its present version this is not possible to fully assess*.

*1) For RNA and Ribo-Seq experiments, there is no information given concerning the depth of sequencing. How many million mapped reads and how many replicates were done for each sample*?

We have included biological replicates for all Ribo-Seq and mRNA-seq data for all conditions. Figure 1–figure supplement 7 contains the reads mapped information and Figure 1–figure supplement 8 shows the correlation coefficient between replicates.

*2) The significant changes in ribosome occupancy are reported in supplementary tables, but changes in RNA transcript abundance are not reported. This data should also be added to the manuscript*.

RNA transcript abundance is shown in Figure 1–figure supplement 9.

*3) For the Ribo-Seq fold changes reported, how are reads mapping to overlapping ORFs handled? Eg*
Figure 1—figure supplement 1*, shows reads covering the uORFs (green) and main ORFs (blue) that overlap*.

We removed Ribo-Seq-fold changes of uORFs from the manuscript.

*4) Is the read density data in*
Figure 1—figure supplement 1
*reporting a single representative sample, total summed reads from replicates (if done) or mean reads from multiple replicates? The read density is not very high for any of the genes shown, except for ATF4*.

The read density represents the total reads of two replicates. We have only included ATF4 and SLC35A4, which have sufficient overall higher read density to draw strong conclusions.

*5) Can the authors confirm and comment on the presence of RPFs between the annotated uORF and main ORF for CHOP mRNA*?

The data have been removed.

*6) One perhaps unexpected finding was an apparent increase in uORF RPFs following stress. The authors have chosen not to elaborate on this point. The usual translational control models depicted often show an all or nothing extreme response, but it is not surprising at all that there are still ribosomes on uORFs that may get skipped more frequently under stress conditions. Perhaps an important measure of control is the proportion of ribosomes that skip uORF2 in ATF4 and 5. This could be quantified as the Ribo-Seq read data for the overlapping ORF regions should be in separate reading frames. It should be possible to assign reads to either the uORF or the main ORF. As a measure of stress mediated translational control, the authors could quantify the number and proportion of RPF within the ORFs key for translational control. For ATF4 and ATF5 ORF/uORF2 reads would give an alternative readout of the relative translational control*.

The data are no longer presented in the manuscript. The analysis of remaining uORF occupancy in the face of downstream ORF translation will require extensive additional studies.

*7) Blots showing eIF2 phosphorylation relative to total eIF2 should be in each type of cell used (plus minus) treatment for the time points used in the Ribo-Seq study. This could be added to the ATF4 blot which uses different time points*.

The ATF4 blot has been removed as it related to the MEFs data. We have reported time courses of eIF2α phosphorylation and ATF4 production in HEK293T cells in Figure 3—figure supplement 1 of our original *eLife* paper (10.7554/eLife.00498). We do not detect phospho-eIF2α phosphorylation or ATF4 production after only 1 hour of Tm treatment by Western blot analysis, which is the condition used for ribo and mRNA-seq analysis. Ribo-Seq analysis is a more sensitive method than Western blotting to detect the early translational changes upon ER-stress.

*8) For the immunofluorescence experiments and for the GFP experiments in*
Figure 3
*and 4 and Figure 4*–*figure supplement 1, each experiment requires quantification of a larger number of cells than are shown in each representative image. For example in 100 cells in each of three separate experiments what proportion of cells (plus minus) error contain stress granules or P bodies as appropriate. Such an analysis would enable statistical treatment of the significance of the observations made*.

The quantitation was added to the stress granule data.

*9) What is the effect of the ISR and ISRIB on the other arms of the UPR at the times used here. Are sampling times too early to observe XBP1 splicing and changes in ribosome binding? If they are this should be stated, if they are not it would helpful to show XBP1 reads*.

At the time points analyzed, it is too early to detect changes in XBP1 mRNA splicing. In Figure 3—figure supplement 1 of our original *eLife* paper (10.7554/eLife.00498), we looked at XBP1s induction upon Tm treatment in HEK293T cells. XBP1s production lags ATF4 translational upregulation and is not detected at 1 hour of Tm treatment. We also reported in our previous manuscript that ISRIB treatment or blocking the PERK branch of the UPR does not alter activation of IRE1 and XBP1 splicing but it prolongs activation of this branch as measured both by IRE1-GFP foci formation and XBP1 splicing (Figure 5C and Figure 5–figure supplement 2).

Reviewer #2

*[…] While the study is potentially interesting, there are some important issues to address*.

*Major issues*:

*Throughout the manuscript, there is a disconnection between the data and the conclusions. I will only highlight the major ones here*:

*1) One of the central conclusions of the paper is that ISRIB eliminated the effects of eIF2*α *phosphorylation. The authors wrote: “The translational output of ISRIB-treated cells to ER stress was remarkably similar to that of cells with a genetically ablated ISR”. This contradicts the data. In fact, there are major differences between the two data sets (*Figure 2*). I recommend providing a complete list of genes that change after ISRIB treatment and to study these changes, as it may shed light on ISRIB's function and/or target*.

We have removed the MEF data and thus we do no longer compare ISR-ablated cells with ISRIB-treated cells. In both the original submitted *eLife* Research Advance and in the revised manuscript, we have provided a list of genes that are significantly and substantially translationally upregulated in all conditions reported (Tm, Tm + ISRIB and ISRIB alone). We have carefully analyzed the genes that remain translationally upregulated in the presence of Tm + ISRIB. Unfortunately they are all hypothetical proteins with unknown functions and thus do not shed light on ISRIB function. Some of these uncharacterized genes are also down regulated in the presence of ISRIB alone.

*Supporting the idea that the changes caused by ISRIB (*Figure 2*) are actually tractable, ISRIB caused more changes (*Figure 2*) than Tm (*Figure 1*). The secret of ISRIB's function may be hidden in this dataset. This needs to be documented and exploited*.

As stated above, ISRIB changes have been documented. Compared to Tm treatment, ISRIB induced changes are small.

*A related issue: The authors conclude that “ISRIB does not have general off-targets effects on translation, transcription or mRNA stability.” Again, this contradicts the data. Beside, “off-target” is a not suitable here, since we don't know what is the target of ISRIB*.

We have removed off-target from the text. The ribosome and mRNA genome-wide profiling data demonstrates that ISRIB comprehensively blocks the translational effects driven by UPR activation and eIF2α phosphorylation and does not lead to overall and spurious changes in mRNA levels or translation.

*2)*
Figure 3*: What are the vertical lines on PERK blots? Between first and second lane in the top blot and between the 9th and 10th lane in the bottom blot*.

All lanes were contiguous in the blots. The film was rescanned and the lines are no longer present.

*3)*
Figure 3*: It would be good to see images with better resolution to appreciate the localization of eIF3α, which looks very interesting. I still see some stress-granules on Ars+ISRIB in some cells but in most cells, the signal is too strong to be analyzed. It would be good to see images of better resolution and some quantitative assessment of the effects*.

Quantitative assessment was included for all stress granule data. Immunofluorescence data of *eIF3α* with all stressors used in this paper has been previously published.

*4) The disappearance of stress-granules is the impressive finding of the manuscript but how does this all happen? It is difficult to follow what might be going on because of the lack of consistency in the conditions used and because adequate controls are not presented. From this paper and the previous one, it would appear that ISRIB acts upstream of ATF4 translation and is dependent upon eIF2*α*-P (but without affecting the latter)? This needs to be explored further to get some understanding of what is happening*.

The goal of using different stressors was to distinguish between phospho-eIF2α-dependent (thapsigargin and arsenite) and independent (panteamine A and hippuristanol) stress granule formation. As expected by the ability of ISRIB to make cells resistant to the effects of eIF2α phosphorylation, the small molecule was able to block only phospho-eIF2α-dependent stress granule formation. ATF4 is downstream of eIF2α phosphorylation and thus it is also blocked by ISRIB. The mechanism of action of the drug will await identification of its molecular target.

*Figure 4C: It would be good to present gels to show the effects, as the authors have the images already. It would be interesting to appreciate the qualitative changes in translation. Does this match, what is seen by ribosome profiling*?

We have included the gel in Figure 3—figure supplement 2. The live cell imaging and ^35^S-methionine incorporation and recovery experiments were performed in U2OS cells expressing G3BP-GFP using the potent ER-stressor, thapsigargin. It is not the same cell type or the same ER-stressor (tunicamycin) used in the ribosome footprinting analysis.

*5) The Discussion is very broad and not connected to the current dataset. The first sentence of the Discussion is incorrect, as there are other antagonists of the ISR. I suggest two options for a revised discussion: Either to focus the discussion on the current dataset or to provide additional data inline with the discussion (mTOR signaling and the ISR, memory and stress granules, memory and ribosome profiling data)*.

We have substantially rewritten the Discussion. To the best of our knowledge however, ISRIB is indeed the first and to date only known ISR inhibitor. Although there are inhibitors of specific eIF2α kinases (PERK and PKR inhibitors), ISRIB is the only molecule that can block signaling downstream of all eIF2α kinases by making cells resistant to eIF2α phosphorylation. An ISR agonist exists, salubrinal, which prolongs eIF2α phosphorylation. An HRI activator also exists which can also act as an agonist of the ISR in cells that express HRI.

*6)*
Figure 1*: I didn't understand this panel. It needs to be clarified*.

This figure was removed.

Reviewer #3

*[…] The lack of any measureable effect of ISRIB on baseline translation is surprising as there is evidence for basal activity of the ISR: For example PERK_KO, eIF2a_S51A and ATF4_KO cells all share a strong baseline requirement for amino acid supplementation (likely indicating that the ISR contributes to baseline ATF4-mediated gene expression). The authors may wish to comment on this point*.

HEK293T cells were only treated with ISRIB for one hour and thus the basal levels of ISR activation may not be as evident as in cells under prolonged ablation of PERK (PERK^-/-^), ATF4 (ATF4^-/-^) or non-phosphorylatable eIF2α (eIF2α ^S51A/S51A^).